# Bacterial adaptation to rhizosphere soil is independent of the selective pressure exerted by the herbicide saflufenacil, through the modulation of catalase and glutathione S-transferase

**Caroline Rosa Silva**[1], **Amanda Flávia da Silva Rovida**[2], **Juliane Gabriele Martins**[1], **Paloma Nathane Nunes de Freitas**[3], **Luiz Ricardo Olchanheski**[1], **Luciana Grange**[4], **Sônia Alvim Veiga Pileggi**[1], **Marcos Pileggi**[1] *

**1** Department of Biological and Health Sciences, Department of Structural, Molecular and Genetic Biology, State University of Ponta Grossa, Ponta Grossa, Paraná, Brazil, **2** Department of Biotechnology, Genetics and Cell Biology, State University of Maringá, Maringá, Brazil, **3** Luiz de Queiroz College of Agriculture, University of São Paulo, Piracicaba, Brazil, **4** Department of Agricultural Sciences, Federal University of Paraná—Palotina Sector, Palotina, Brazil

* mpileggi@uepg.br

## Abstract

Herbicides cause oxidative stress in nontarget microorganisms, which may exhibit adaptive responses to substances they have not previously encountered. Nevertheless, it is unclear whether these characteristics occur in bacteria isolated from agricultural soil. Two possible adaptation strategies of *Stenotrophomonas* sp. CMA26 was evaluated in agricultural soil in Brazil, which is considered stressful due to the intense use of pesticides. The study focused on degradation and antioxidant enzymes in response to the herbicide Heat, which was absent at the isolation site. The results indicated that higher concentrations of herbicide led to more intense stress conditions during the initial periods of growth. This was evidenced by elevated levels of malondialdehyde and peroxide, as well as a significant reduction in growth. Our data show that herbicide degradation is a selection-dependent process, as none of the 35 isolates from the same environment in our collection were able to degrade the herbicide. The stress was controlled by changes in the enzymatic modulation of catalase activity in response to peroxide and glutathione S-transferase activity in response to malondialdehyde, especially at higher herbicide concentrations. This modulation pattern is related to the bacterial growth phases and herbicide concentration, with a specific recovery response observed during the mid phase for higher herbicide concentrations. The metabolic systems that contributed to tolerance did not depend on the specific prior selection of saflufenacil. Instead, they were related to general stress responses, regardless of the stress-generating substance. This system may have evolved in response to reactive oxygen species, regardless of the substance that caused oxidative stress, by modulating of the activities of various antioxidant enzymes. Bacterial communities possessing these plastic tolerance mechanisms can survive without necessarily degrading herbicides. However, their presence

**Data Availability Statement:** The bacterial strain was identified as belonging to the genus Stenotrophomonas, accession number OL631162 - NCBI, named Stenotrophomonas sp. CMA26.

**Funding:** This work was supported by the Coordination for the Improvement of Higher Level Personnel (CAPES) by granting a scholarship.The funders had no role in study design, data collection and analysis, decision to publish, or preparation of the manuscript.

**Competing interests:** The authors have declared that no competing interests exist.

can lead to changes in biodiversity, compromise the functionality of agricultural soils, and contribute to environmental contamination through drift.

## Introduction

Agriculture relies on the use of pesticides for weed control in order to maintain crop yields and sustain the growth of the human population. However, the indiscriminate use of these substances impacts the environment, primarily by affecting non-target species and promoting the growth of resistant weeds [1]. This effect has led farmers to use larger amounts and varieties of herbicides to control them [2].

Heat herbicide is a commercial brand in Brazil that contains the active molecule saflufenacil. It is used as an alternative to control glyphosate-resistant weeds [3]. An increase in the use of this herbicide is anticipated due to the increasing regulatory pressures on glyphosate in the coming years [4]. The mode of action of Heat herbicide is through the inhibition of protoporphyrinogen oxidase (Protox) [5]. However, this target is not exclusive to weeds, as it is also present in several prokaryotes such as Firmicutes, *Escherichia coli*, *Salmonella*, and some cyanobacteria [1]. The saflufenacil molecule still contains the chemical elements fluorine and chlorine, which belong to the group of halogens. These elements give the herbicide high oxidizing activity due to the electronegativity of the group [6]. In light of these factors, the Heat herbicide has a high potential for affecting non-target microorganisms.

Herbicides promote oxidative stress in microorganisms, causing an imbalance between the production and elimination of reactive oxygen species (ROS). This imbalance leads to cellular metabolic disorders [7, 8]. The herbicide mesotrione, for example, promoted oxidative stress and slowed population growth rates in *E. coli* DH5-$\alpha$ [9]. It also induced significant increases in hydrogen peroxide ($H_2O_2$) production in *Pantoea ananatis* [10]. High levels of peroxide were observed in *Pseudomonas* sp. CMA-7.3 on exposure to the herbicide 2,4-dichlorophenoxyacetic acid (2,4-D) [11].

Microorganisms have antioxidant systems that employ both non-enzymatic and enzymatic responses to eliminate or minimize the damage caused by ROS [12]. The enzymes that have been extensively studied for ROS control are superoxide dismutase (SOD), which catalyzes the dismutation of superoxide ($O_2^-$) into $H_2O_2$ and oxygen ($O_2$), and catalase (CAT), which breaks down $H_2O_2$ into water ($H_2O$) and $O_2$ [5]. Glutathione S-transferase (GST), which acts in the detoxification of xenobiotics by using glutathione (GSH) to form conjugates, also plays a role in redox metabolism. It protects the cell from damage caused by oxidative stress [13]. Gene expression analysis of *Stenotrophomonas maltophilia* showed that the enzymes KatA2 and alkyl hydroperoxidase were responsible for the survival of the bacteria when exposed directly to high concentrations of $H_2O_2$ [14]. Increases in GST activity, as well as GSH concentration, occurred in *Pseudomonas fulva* 4C07 exposed to the herbicide clomazone, either alone or in combination with ametryn [15]. Increases in GST activity and a polymorphic CAT system efficiently controlled the oxidative stress caused by mesotrione in *P. ananatis* [10].

Microorganisms with resistant or tolerant phenotypes are positively selected in ecosystems that experience strong selective pressure due to the intensive use of pesticides, such as agricultural soils [16]. Decreases in the diversity of nitrogen-fixing and nitrifying bacteria, as well as increases in denitrifying bacteria, were observed in soils exposed to mesotrione [17] and clomazone [18]. Microorganisms belonging to the phylum Acidobacteria, which are involved in biogeochemical processes, showed a decrease in abundance in the rhizosphere of corn and

soybean plants after prolonged glyphosate application [19]. In this context, prior selective pressure plays a relevant role in bacterial adaptation [20], such as the development of herbicide degradation phenotypes. This is often observed in microorganisms that have been previously selected for their ability to metabolize xenobiotic agents [21].However, some studies have shown that bacteria can survive in pesticide-contaminated environments even without pre-existing selective pressure. The response system related to this adaptation is based on general stress responses characterized by the bacterial cell's ability to defend itself not only from the specific stress-inducing factors but also from other seemingly unrelated factors [22]. The formation of biofilms to protect cells against the herbicides Aminol, Atectra, Boral, and Heat [23]; cross-resistance by efflux pumps that export glyphosate and antibiotics [24]; increased gene expression and copy number of genes to tolerate pesticide stress [25]; horizontal transfer of genes that confer multi-drug resistant phenotypes [26], and the antioxidant system are examples of general stress response mechanisms. Mutants with deletions in genes encoding Mn-SOD (*sod*A) and Fe-SOD (*sod*B) of *E. coli* K-12, a non-environmental strain not previously exposed to any pesticide, showed tolerance to the Gramoxone herbicide through the activities of isoforms of antioxidant enzymes CAT and SOD [27]. *Pseudomonas* sp., isolated from biofilm formed in tanks used for washing pesticide containers, exhibited a mechanism of regulation of antioxidant enzyme activities to enhance tolerance to high concentrations of Boral herbicide. This herbicide was not present at the site where the bacteria were isolated [28].

It is still unclear whether bacteria isolated from agricultural soil may have survival mechanisms that are independent of previous selection for specific xenobiotics. Thus, this work investigates whether *Stenotrophomonas* sp. CMA26, a Heat herbicide-tolerant strain without a history of previous exposure to this herbicide, may have response mechanisms to survive at concentrations higher than those used in agriculture.

## Material and methods

The experimental design is shown in S1 Appendix. The *Stenotrophomonas* sp. CMA26, a strain from the Microorganisms Collection at the Environmental Microbiology Laboratory of the State University of Ponta Grossa (UEPG) in Ponta Grossa, Paraná, Brazil, was assessed for its tolerance to the Heat herbicide. The study also examined stress indicators such as the quantification of $H_2O_2$ and malondialdehyde (MDA), as well as the mechanisms of response to the Heat herbicide, including degradation capacity and the activities of CAT and GST enzymes, using concentrations higher than those typically used in agriculture.

### Chemicals and culture mediums

The herbicide used in this study was Heat (BASF—Ludwigshafen, Rhein, Germany), which containing 700 g/L (70% w/v) of the active ingredient saflufenacil (N'-{2-chloro-4-fluor-5-[1,2,3,6-tetrahydro-3-methyl-2,6-dioxo-4-(trifluoromethyl)pyrimidin-1-yl]benzoyl}-N-isopropyl-N-methylsulfamide).

The recommended field dose of Heat herbicide is equivalent to a concentration of 0.49 mM of saflufenacil (1x).

Broth Luria Bertani (LB: 10 g/L tryptone; 10 g/L NaCl; 5 g/L yeast extract, pH 7.0 ± 0.2) was used for bacterial culture. The concentrations of the Heat herbicide used in this study were as follows: LB without herbicide (C); LB + 0.49 mM saflufenacil (1x); LB + 4.9 mM saflufenacil (10x); LB + 24.5 mM saflufenacil (50x). Temperature and agitation were standardized at 30°C and 4 g, respectively. Assays were performed in triplicate.

## Isolation and collection and of microorganisms

From rhizosphere soil, 10 g were collected from four agricultural areas for soybean and corn cultures in the city of Palotina, Paraná, Brazil, using four parallel transepts (S2 Appendix). The pesticides used in these agriculture areas and their chemical characteristics are shown in S3 Appendix. Heat herbicide was not used at the bacterial isolation sites.

The collected soil was homogenized in 100-mL PBS (pH 6.8, phosphate buffered saline: 8 g $L^{-1}$ NaCl; 0.2 g $L^{-1}$ KCl; 1.44 g $L^{-1}$ Na$_2$HPO$_4$; 0.24 g $L^{-1}$ KH$_2$PO$_4$). 100 μL were sequentially diluted and inoculated in LA (LB + agar) and incubated, as described above. Pure colonies were isolated. The 35 bacterial isolates, including the strain *Stenotrophomonas* sp. CMA26, were added to the Environmental Microbiology Laboratory Collection and maintained in glycerol at -80°C. The microorganisms were collected by Dr. Luciana Grange, one of the authors of this work.

## Heat tolerance test

Thirty-five bacterial isolates were used to evaluate the tolerance to Heat herbicide. The isolates were inoculated in 96-well plates with 100 μL of culture medium added in each well. Growth rate was evaluated after 24 h by optical density (OD) in a microplate reader at 600 nm.

## Molecular identification of bacterial strain by 16S ribosomal gene sequencing

The *Stenotrophomonas* sp. CMA26 was cultivated in LB for 24 h and incubated under the conditions as described before. Subsequently, 200 μL of the cultured were collected and subjected to molecular identification by sequencing the 16S rRNA gene. Total DNA was extracted using the ReliaPrep™ Blood gDNA Miniprep System Kit—Promega (Madison, WI, USA). The primers fD1(5′ -CCGAATTCGTCGACAACAGAGTTTGATCCTGGCTCAG-3′) and rD1(5′ - CCCGGGATCCAAGCTTAAGGAGGTGATCCAGCC-3′) were used for the amplification. The reaction (PCR) consisted of an early cycle of denaturation at 95°C for 5 min; 30 cycles of denaturation at 94°C for 45 s, annealing at 55°C for 45 s and extension at 72°C for 2 min; and a final extension cycle for 10 min at 72°C. PCR products were purified using Qiagen (Hilden, Ger), QIAquick PCR kit (n° 28104). After purification of the PCR products, band integrity was checked through electrophoresis, and the material obtained was sent for sequencing at Ludwig Biotec (Alvorada, Brazil). The sequences were analyzed using resources from the Ribosomal Database Project site. The strain was identified as belonging to the genus *Stenotrophomonas*, accession number OL631162—NCBI, named *Stenotrophomonas* sp. CMA26.

## Bacterial growth curve

The *Stenotrophomonas* sp. CMA26 pre-inoculum was prepared in 100-mL culture medium, until it presented an OD of approximately 2.0 at 600 nm, in a spectrophotometer. The inoculum was transferred to treatment cultures at an initial OD of 0.05 at 600 nm and incubated as described above. The OD was monitored every hour in a spectrophotometer, until the beginning of the stationary phase, and the data were plotted on a growth curve. For higher precision in the quantification of bacterial growth, dilutions with LB culture medium were performed whenever the OD was greater than 1.0.

## Cell viability

The *Stenotrophomonas* sp. CMA26 was grown in 100-mL LB, as described before. Samples of 100 μL were collected at 4, 6, and 8 h of incubation and sequentially diluted to $10^{-6}$, inoculated

into LA and incubated as described. Colony forming units (CFU) were counted after 24 h of incubation.

## Pre-treatment of samples for evaluation of oxidative stress

The *Stenotrophomonas* sp. CMA26 was cultured in 100-mL culture medium as described before, and the cells were centrifuged at 8,000 *g* for 10 min at 4°C at 4, 6, and 8 h of incubation. The supernatant was discarded, and the precipitate was macerated in liquid nitrogen. Aliquots of 100 mg of the precipitate were frozen at -80°C until analysis.

## Quantification of $H_2O_2$

Extracted samples, already described, were homogenized with 1 mL of 0.1% trichloroacetic acid (TCA) and centrifuged at 10,000 *g* for 15 min at 4°C for quantification of $H_2O_2$ [29]. Subsequently, 0.2 mL of the supernatant was collected and transferred to a tube containing 0.2 mL of a 100 mM phosphate buffer solution (14.52 g/L $K_2HPO_4$, 2.26 g/L $KH_2PO_4$), pH 7.5 and 0.8 mL of 1M potassium iodide (KI) solution. The sample was stored in the dark and on ice for 1 h. The reading was performed in a spectrophotometer at 390 nm. Results were expressed as µmol $H_2O_2$ $g^{-1}$ fresh mass.

## Quantification of MDA

The extracted samples, already described, were homogenized with 1 mL of 0.1% TCA and centrifuged at 10,000 g for 5 min [30]. A 0.25 mL aliquot of the supernatant was collected and transferred to a tube containing 1 mL of a 20% TCA + 0.5% 2-thiobarbituric acid (TBA) solution. The mixture was incubated in a water bath at 97°C for 30 min and cooled on ice for 10 min. The sample was centrifuged at 10,000 g for 10 min, and the supernatant was analyzed in a spectrophotometer at 535 nm (wavelength for reading total lipids) and 600 nm (wavelength for reading total lipids, minus MDA), respectively. The amount of MDA was expressed in mmol/mg fresh mass.

## Degradation capacity of bacterial isolates

The 35 bacterial isolates were cultivated in LB. After 24 h, cells were centrifuged at 8,000 g for 5 min at 4°C. The supernatant was discarded, and the concentrated cells were resuspended in 10-ml LB. The equivalent of the recommended dose for field use of herbicide Heat (1x) was added to the sample, and the cells were incubated again. Aliquots of 1 mL were collected from the culture medium at 0 and 24 h of incubation, and centrifuged at 12,000 g for 5 min. The supernatant was frozen. Samples were filter-sterilized before being injected into liquid chromatography coupled to mass spectrometry (LC-MS/MS) equipment. The experiments were conducted in triplicates, comparing the samples at 0 and 24 h, to observe the decrease in the amount of herbicide after the incubation time.

## Analysis in LC-MS/MS

The analysis of LC-MS/MS were carried out with the injection of 10 µL of the prepared extracts (already described), at a flow rate of 0.2 mL min$^{-1}$ in the LC-MS/MS model Acquity and Xevo TQD (Waters Corporation, Milford, Massachusetts, USA) equipped with binary pumps, a vacuum degasser, a SIL-HTc autosampler, using an ACQUITY UPLC® BEH C18 2.1 x 50 mm column (1.7 µm porosity) maintained at 50°C. The UPLC system consisted of 0.1% formic acid (mobile phase A) in ultrapure water and a mobile phase B made of standard HPLC acetonitrile. The gradient profile was as follows: isocratic over 1 min, a linear gradient from 10% to

90% B over 4 min, followed by 100% B over 1 min, with a flow rate of 0.2 mL/min$^{-1}$. The column was re-equilibrated for 2 min. The MS experiment was conducted on a triple quadrupole 4000 linear ionic traction mass spectrometer from QTRAP (Applied Biosystem, Foster City, CA, USA) equipped with a Turbo-Ion source. The instrument was operated in the negative electrospray ionization (ESI) mode and data were acquired in the multiple reaction monitoring (MRM) mode. The conditions of the MS experiment were as follows: capillary voltage set at 2.9 kV and cone voltage at 55 V; the source temperature and the desolvation temperature maintained at 150˚C and 350˚C, respectively. The desolvation gas was set at a flow of 650 L h$^{-1}$; 499.2 (m/z) was selected as the precursor ion and its quantitative and qualitative ions were 348 (m/z) and 328 (m/z), respectively; when the collision energies are 32 V and 41 V, respectively. For UPLC analysis, Masslynx NT v.4.1 (Waters) software was used to process data. Under the conditions described the retention time of saflufenacil was approximately 5.30 min.

## Protein extraction and quantification

The *Stenotrophomonas* sp. CMA26 was grown in 100 mL culture medium to obtain protein extracts, as described before. Cells were centrifuged at 8,000 *g* for 10 min at 4˚C, at 4, 6, and 8 h of incubation, according to the bacterial growth curve and representative of growth phases studied in this work. The supernatant was discarded, and the precipitate was macerated in liquid nitrogen and homogenized in 100 mM potassium phosphate buffer pH 7.5 containing 1 mM ethylene diamine tetraacetic acid (EDTA), 3 mM dithiothreitol (DTT) and 5% polyvinylpyrrolidone (PVPP). The extract was centrifuged at 10,000 *g* for 30 min, and aliquots of the supernatant were stored and frozen for further analysis.

The total protein concentration was determined bovine serum albumin (BSA) as a standard [31]. Results were expressed as μmol protein g$^{-1}$ fresh mass.

## CAT activity

CAT activity was determined at 25˚C, in a solution containing 1 mL of 100 mM potassium phosphate buffer (pH 7.5) and 2.5 μL of $H_2O_2$ (30%) [32]. The reaction was started with the addition of 25 μL of protein extract, as described, and the activity was determined through the decomposition of $H_2O_2$ and monitored in a spectrophotometer at 240 nm for 1 min. Results are expressed in μmol/min/mg prot.

## GST activity

GST activity was determined through the assay with a solution containing 900 μL of 100 mM potassium phosphate buffer (pH 6.8), 25 μL of 40 mM 1-chloro-2,4-dinitrobenzene (CDNB) and 50 μL of GSH 0.1 mM and incubated at 30˚C [33]. The reaction was started with the addition of 25 μL of protein extract and was monitored spectrophotometrically for 2 min at 340 nm. The activity was expressed as μmol/min/mg prot.

## Statistical analysis

Statistical analyses were performed using the Tukey test using the significance criterion $p < 0.05$. Data from quantification of $H_2O_2$ and MDA and from CAT and GST activities were processed by Principal Component Analysis (PCA). The analysis was obtained using the R software version 3.5.1, using the "ggplot2" and "factorextra" data packages.

## Results

### Strain selection for response system studies

Once all the bacterial isolates were evaluated and found to be tolerant to all concentrations of the herbicide Heat, *Stenotrophomonas* sp. CMA26 was selected due to an interesting characteristic of presenting significantly lower growth rates in the early periods of the log phase in the 50x treatment. It was identified by 16S ribosomal gene sequencing.

### Bacterial growth curve

The growth kinetics of *Stenotrophomonas* sp. CMA26, in both the control group and the treatment group with the herbicide Heat, is shown in Fig 1. There are no significant differences in growth rates between the control and treatment groups, except for the lower growth observed during the period of 2 to 6 h in the 50x treatment. This period represents the beginning to the middle of the logarithmic growth phase (log phase) (S4 Appendix). Despite this, after the 6h period of incubation, there are no more statistical differences among all treatments (S4 Appendix). To study the adaptation mechanisms of *Stenotrophomonas* sp. CMA26 to Heat herbicide during periods of stress and recovery. Data on stress indicators and response systems were collected at times of 4 h, 6 h and 8 h intervals, which were classified as early, early-mid, and mid phases, respectively.

### Cell viability

Cell viability data observed in *Stenotrophomonas* sp. CMA26 is shown in S5 Appendix. During the early, early-mid, and mid phases, no significant differences were observed among the

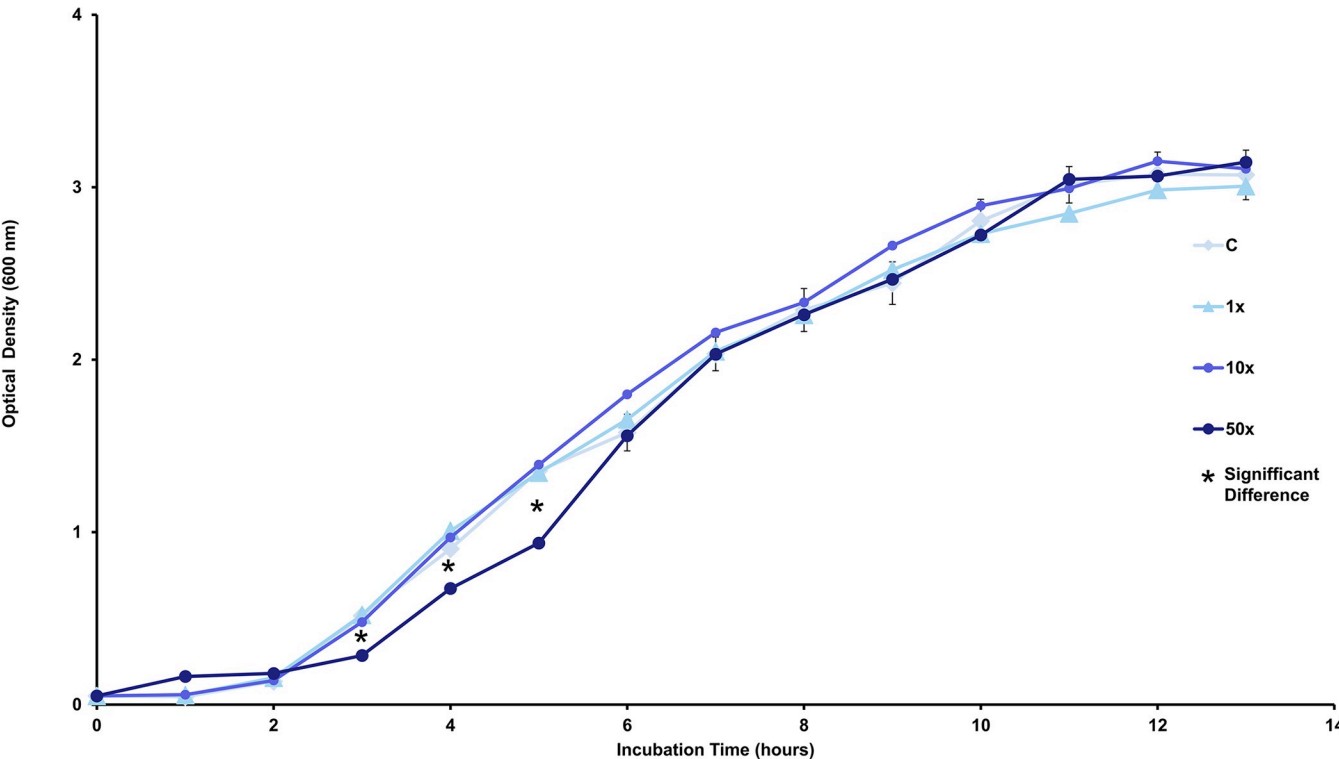

**Fig 1. Growth curve.** Growth curve of *Stenotrophomonas* sp. CMA26 in control (C) and treatments containing 1x, 10x, and 50x the concentration equivalent to that used in the field of herbicide Heat (1x, 10x, and 50x). Asterisks mark the points where statistically significant differences were identified (*). The bars represent the standard errors in the means. Tukey's test (p < 0.05).

control groups (S4 Appendix). The 1x treatment showed significantly higher cell viability than the other treatments in the early phase, but a significant decrease in the early-mid and mid phases. No significant differences were observed among treatments in the early-mid phase. There is a significant increase in cell viability for the 50x treatment compared to the control in the mid phase.

### Indicators of oxidative stress

**H2O2 levels in response to Heat.** The levels of $H_2O_2$ observed in *Stenotrophomonas* sp. CMA26 is shown in Fig 2. $H_2O_2$ levels in the Heat herbicide treatments do not significantly differ from the control in the early phase (S4 Appendix), but there is a tendency for $H_2O_2$ levels to increase in line with the increases in herbicide concentrations. At the 50x treatment, with the highest $H_2O_2$ level in this phase, a drop in the growth rate was observed (Fig 1). The levels of $H_2O_2$ in the control and 1x treatments were significantly lower than those in the 10x and 50x treatments during the early-mid phase. However, there was a reversal of this situation during the mid phase, indicating two distinct patterns of results for $H_2O_2$ levels and herbicide concentrations: one for the control and 1x treatments, and another for the 10x and 50x

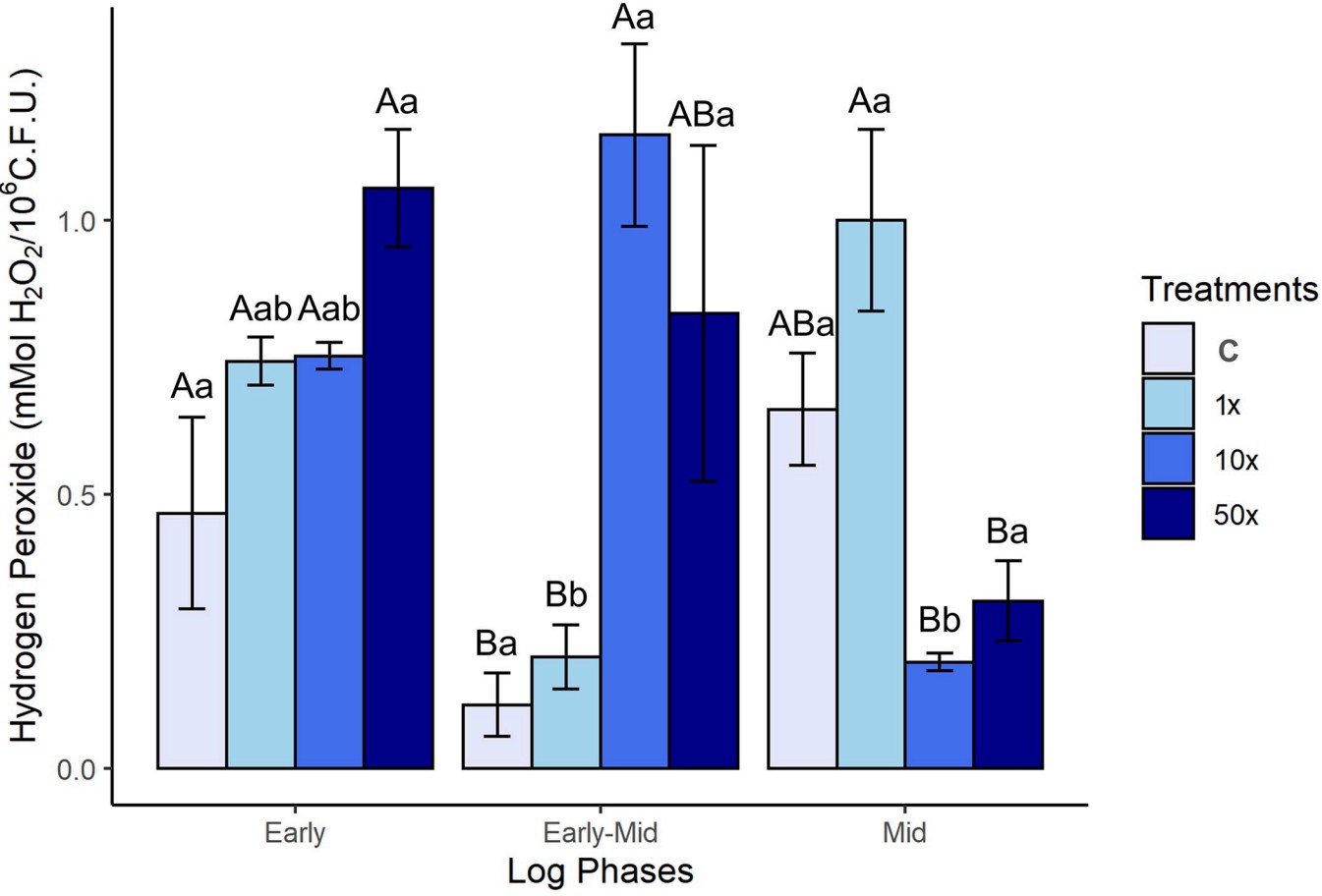

**Fig 2. $H_2O_2$ levels.** $H_2O_2$ levels in *Stenotrophomonas* sp. CMA26 in control (C) and in treatments containing 1x, 10x, and 50x the concentration equivalent to that used in the field of herbicide Heat (1x, 10x, and 50x), in the early, early-mid and mid phases of the log. Uppercase letters statistically compare different treatments from the same growth phase; lowercase letters statistically compare the same treatments at different growth stages. The bars represent the standard errors in the means. Tukey's test ($p < 0.05$).

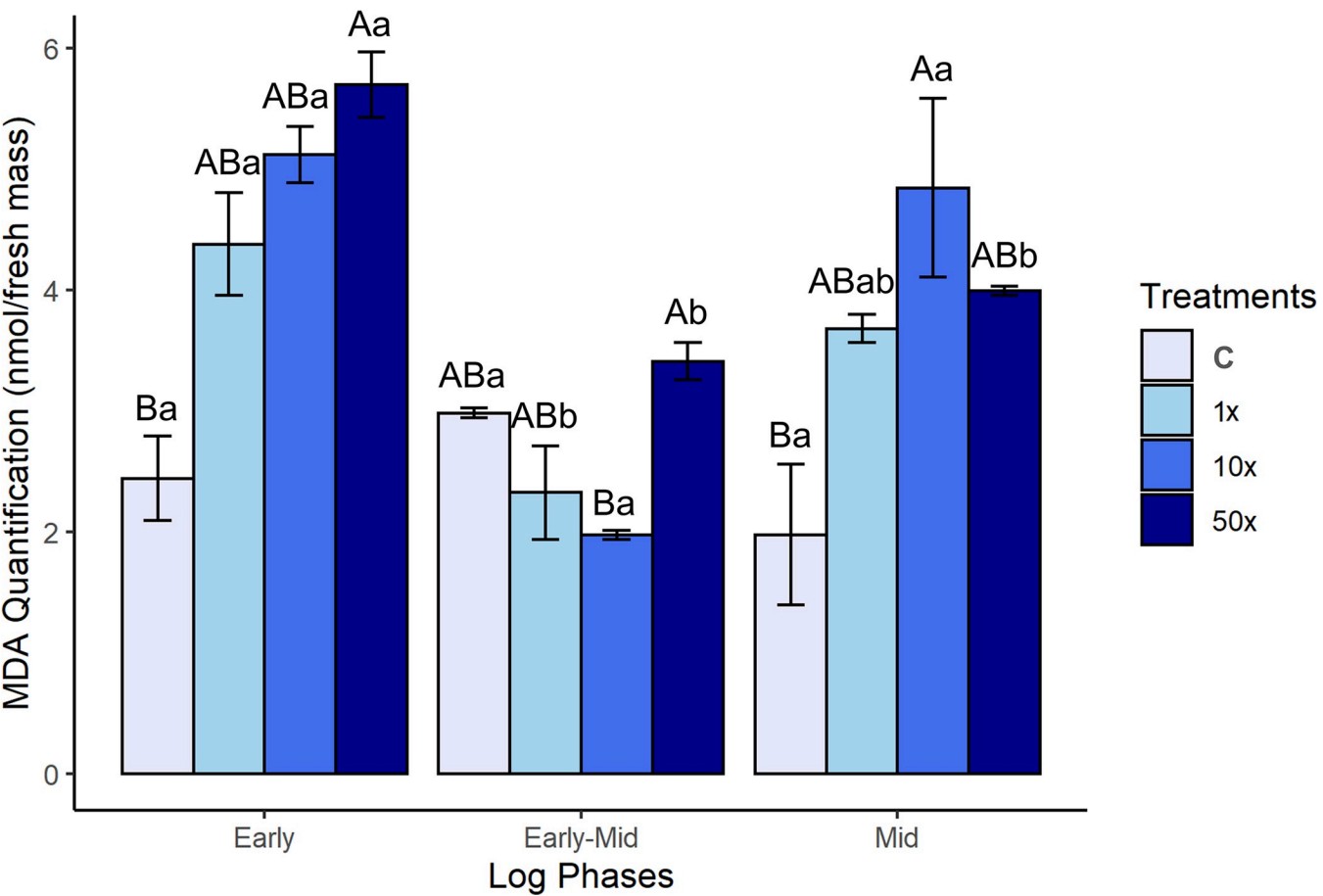

**Fig 3. MDA levels.** MDA levels from *Stenotrophomonas* sp. CMA26 in control (C) and in treatments containing 1x, 10x, and 50x the concentration equivalent to that used in the field of herbicide Heat (1x, 10x, and 50x), in the early, early-mid and mid phases of the log. Uppercase letters statistically compare different treatments from the same growth phase; lowercase letters statistically compare the same treatments at different growth stages. The bars represent the standard errors in the means. Tukey's test (p < 0.05).

treatments. Consistently, this latter phase corresponds to the growth recovery period of *Stenotrophomonas* sp. CMA26 for the 50x treatment (Fig 1).

**Lipid peroxidation in response to Heat.** The MDA levels observed in *Stenotrophomonas* sp. CMA26 are shown in Fig 3. MDA levels did not significantly differ among controls during the early, early-mid and mid phases, but the treatments were significantly higher in the early and middle phases (S4 Appendix). However, the amounts of MDA among controls and treatments were statistically similar in the early-mid phase, but lower than those observed for the other phases. The levels of MDA were directly correlated with the levels of $H_2O_2$ only for the early phase (Fig 2).

## Response systems

**Degradation capacity of bacterial isolates.** None of the bacterial isolates from the collection of microorganisms from agricultural soil showed saflufenacil degradation capacity, including the strain *Stenotrophomonas* sp. CMA26, according to the chromatograms obtained in LC-MS/MS of samples collected at 0 and 24 h of incubation with the herbicide.

**CAT enzymatic response to Heat.** CAT activity in *Stenotrophomonas* sp. CMA26 is shown in Fig 4. The lowest level of CAT activity was observed in the early phase, in which the

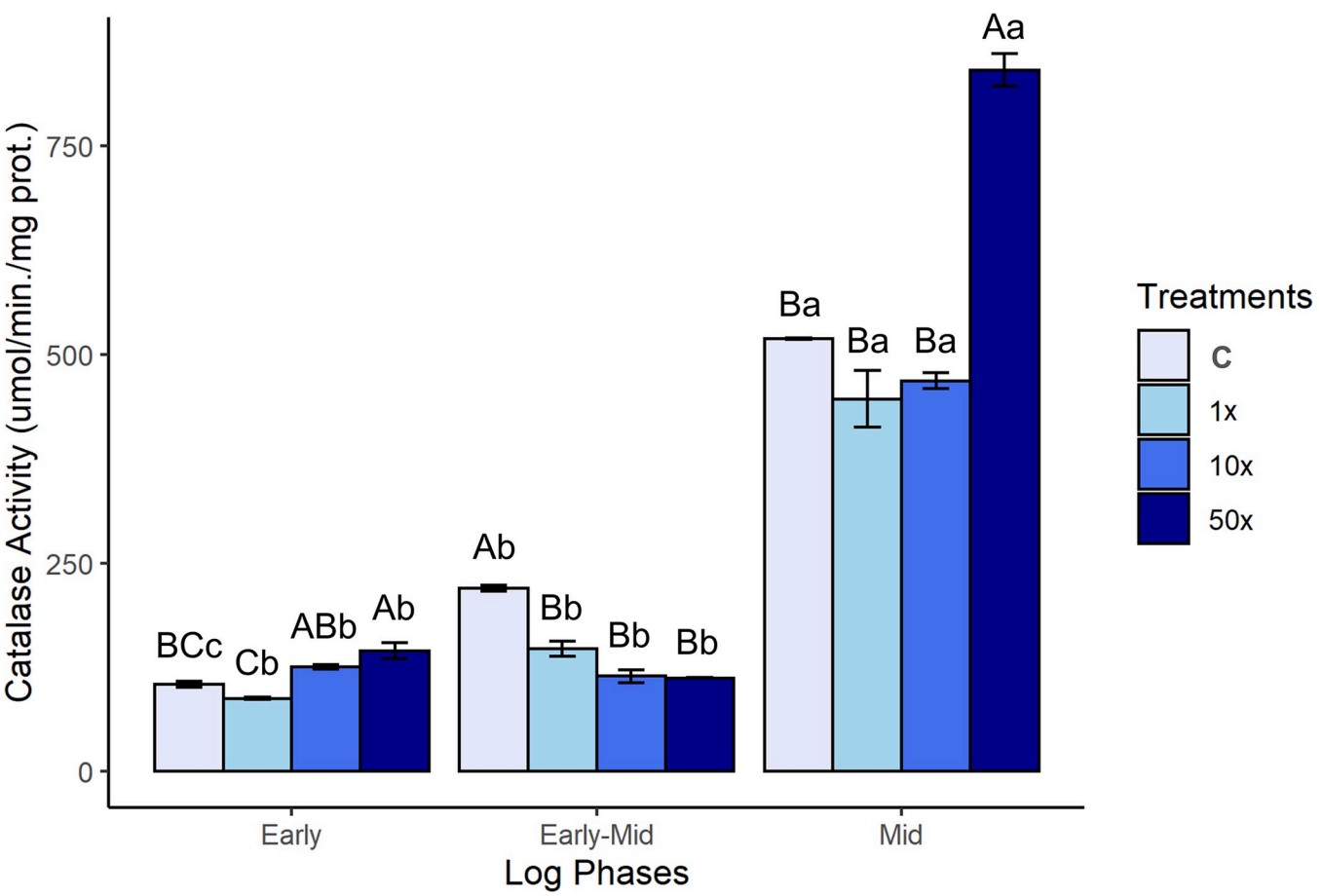

**Fig 4. CAT activity.** CAT activity in *Stenotrophomonas* sp. CMA26 in control (C) and in treatments containing 1x, 10x, and 50x the concentration equivalent to that used in the field of herbicide Heat (1x, 10x, and 50x), in the early, early-mid and mid phases of the log. Uppercase letters statistically compare different treatments from the same growth phase; lowercase letters statistically compare the same treatments at different growth stages. The bars represent the standard errors in the means. Tukey's test ($p < 0.05$).

$H_2O_2$ remains high in all treatments (Fig 2). A significant increase in CAT activity can be seen in control (S4 Appendix) and a reduction in $H_2O_2$ for the treatment in the early-mid phase (Fig 2). There is a significant increase in CAT activity for all treatments in the mid phase. The 50x treatment stands out, with the highest activity of the enzyme (Fig 4) and the reduction of $H_2O_2$ levels (Fig 2) occurring during the period of recovery of growth curve (Fig 1).

**GST enzymatic activity in response to Heat.** GST activity in *Stenotrophomonas* sp. CMA26 is shown in Fig 5, and there were no significantly differences among the controls (S4 Appendix) in the early, early-mid and mid phases. The same was observed for MDA levels (Fig 3). There were significant increases in GST activity from the early-mid to mid phase in treatments, proportional to mid-phase MDA levels (Fig 3).

**Integrated response systems for Heat tolerance.** PCA was used to assess interrelationships among data on oxidative stress indicators $H_2O_2$ and MDA, and enzymatic activities, CAT, and GST (Fig 6). In general, CAT and GST are not associated with any treatment in the early phase. $H_2O_2$ and CAT were distributed in opposite quadrants, first and third, in the early-mid phase; and second and fourth quadrants, respectively, for the mid phase. GST and MDA were distributed in the same quadrants, second, in the early-mid phase; and first quadrant in the mid-phase.

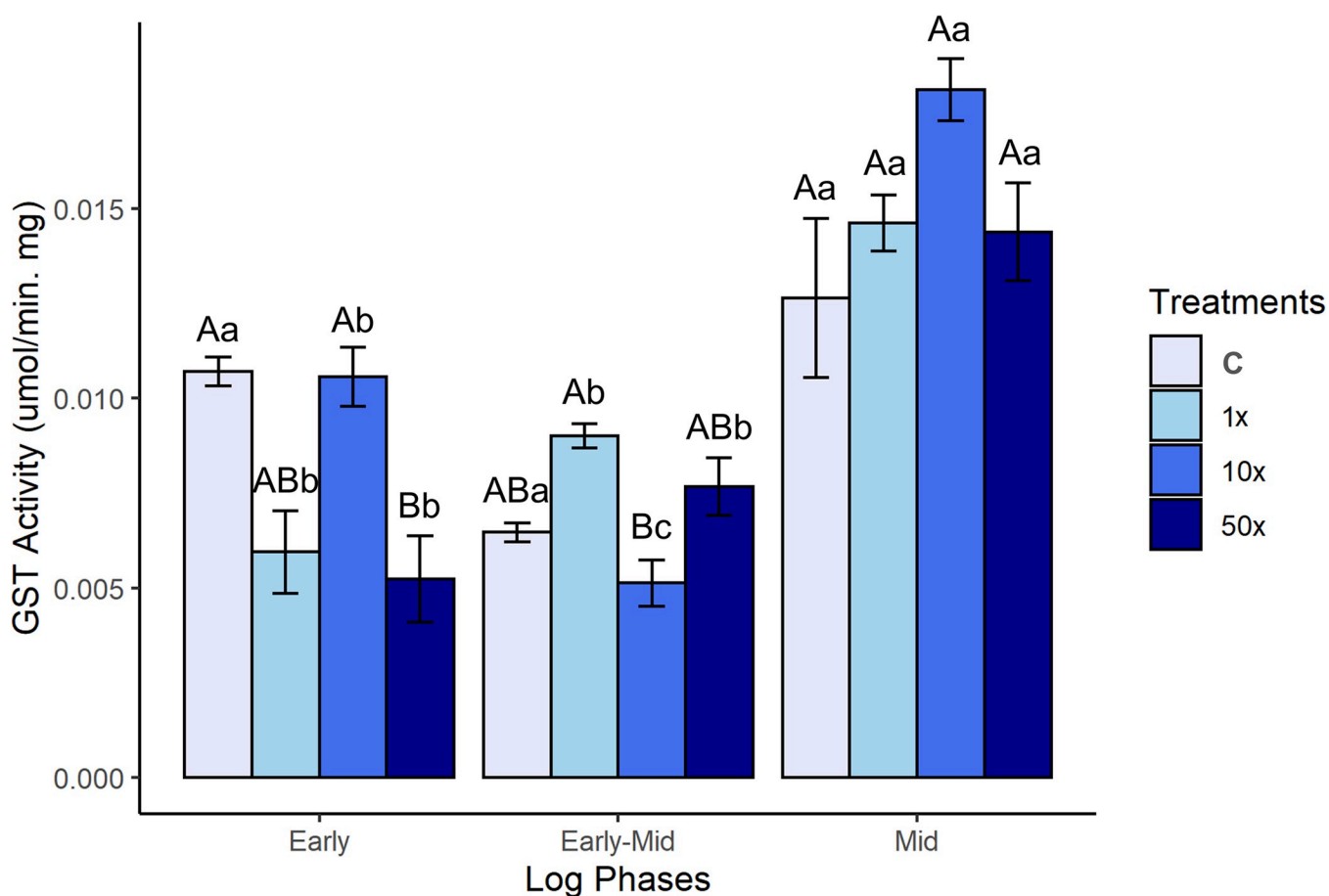

**Fig 5. GST activity.** GST activity in *Stenotrophomonas* sp. CMA26 in control (C) and in treatments containing 1x, 10x, and 50x the concentration equivalent to that used in the field of herbicide Heat (1x, 10x, and 50x), in the early, early-mid and mid phases of the log. Uppercase letters statistically compare different treatments from the same growth phase; lowercase letters statistically compare the same treatments at different growth stages. The bars represent the standard errors in the means. Tukey's test ($p < 0.05$).

Two patterns of responses could be identified: one for control and 1x treatments, and another for 10x and 50x treatments. This distinction was based on the main component that made the greatest contribution in all phases in PC1 (Fig 7). In the early phase, control and 1x treatment were inversely proportional to CAT levels, while 10x and 50x treatments were associated with increased levels of $H_2O_2$ and MDA, which are stress indicators. In the early-mid phase, control and 1x treatments are directly proportional to CAT levels, while $H_2O_2$ levels are related to the 10x and 50x treatments. Finally, in the mid phase, control and 1x are related to $H_2O_2$. Treatments 10 and 50x are related to CAT, GST, and MDA.

## Discussion

### Bacterial tolerance to herbicide

High concentrations of the herbicide Heat during early stages of logarithmic growth are toxic for *Stenotrophomonas* sp. CMA26, according to growth curve in the 50x treatment (Fig 1). The toxicity of herbicide on bacteria has been demonstrated in other studies, such as glyphosate, which can inhibit the growth of bacteria *Rhizobium* sp., *Burkholderia* sp., and *Pseudomonas* sp., isolated from soybean rhizosphere and pastures [34]. The herbicide trifluralin decreased

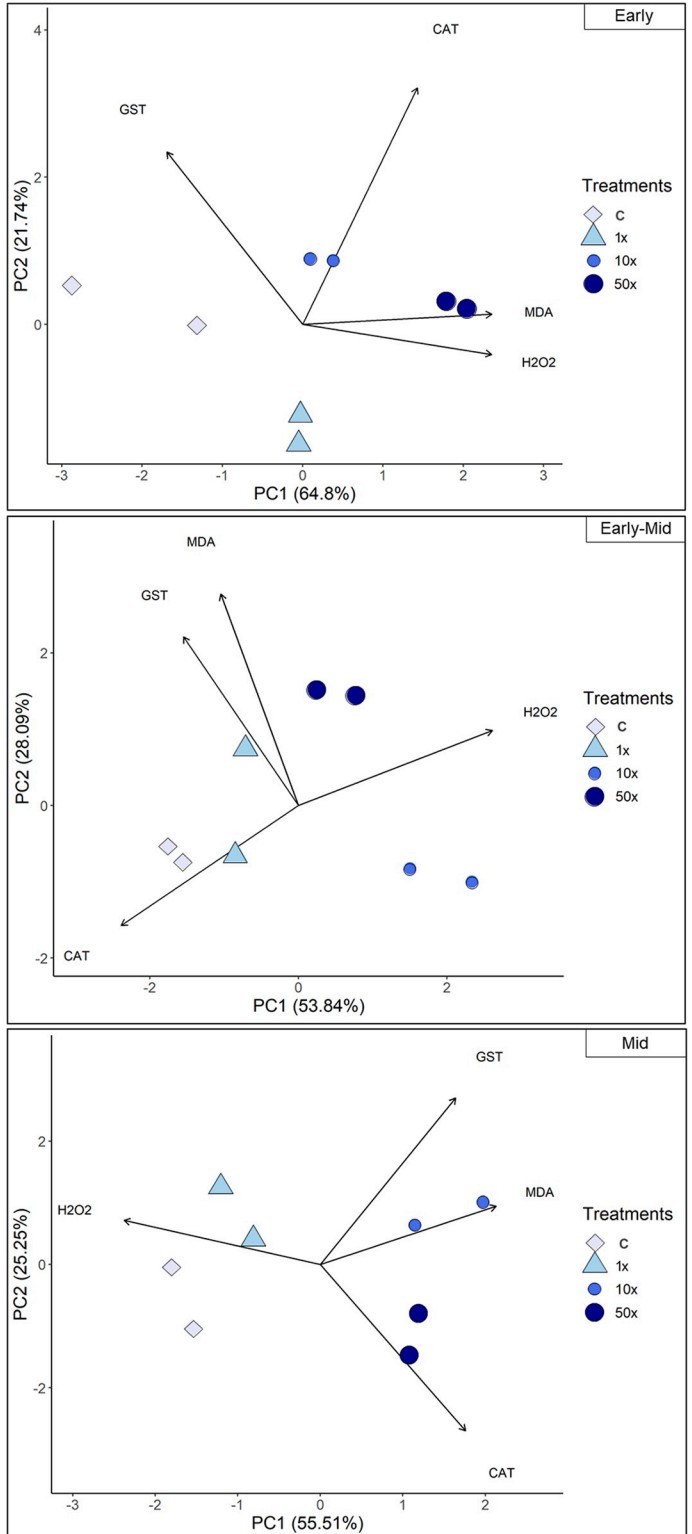

**Fig 6. Relationships between indicators of stress and activities of antioxidant enzymes.** PCA of the relationships between indicators of stress hydrogen peroxide (H$_2$O$_2$), and malondialdehyde (MDA) and profiles of enzymatic activities of catalase (CAT), and glutathione S-transferase (GST) of the strain *Stenotrophomonas* sp. CMA26 in control (C) and treatments containing 1x, 10x, and 50x the concentration equivalent to that used in the field of herbicide Heat (1x, 10x, and 50x), in the early (top panel), early-mid (middle panel) and mid phases (bottom phase) of the log. The

explanation percentages are PC1 = 62.8 and PC2 = 21.74 (early); PC1 = 53.84 and PC2 = 28.09 (early-mid); PC1 = 55.51 and PC2 = 25.25 (mid).

the diversity of nitrogen-fixing and ammonia-oxidizing microorganisms in agricultural soil [35].

The growth recovery of *Stenotrophomonas* sp. CMA26, observed in the early-mid phase at a 50x concentration (Fig 1), seems to indicate the activation of an adaptive response system. Bacteria can exhibit adaptive cellular changes in response to variations in culture conditions that occur during different growth periods, such as alterations in membrane lipid composition. *E. coli*, for example, showed these changes possibly in response to culture osmolarity [36], while *P. ananatis* exhibited changes in response to exposure to the herbicide Calistto and its active molecule mesotrione [10].

*Stenotrophomonas* sp. CMA26 was considered tolerant to Heat herbicide due to its general growth kinetics, even without previous exposure to this herbicide in the site of isolation. The literature, however, focuses on selective processes for bacterial adaptation to herbicides through modifications in community diversity [20]. The phyla Firmicutes and Actinobacteria, which are tolerant to the herbicide nicosulfuron, showed an increase in both diversity and abundance in soil that was exposed to this herbicide [37]. Two Actinobacteria strains isolated from soils treated with Granstar were characterized as tolerant to this herbicide [38]. DNA sequence analysis obtained from the alignable tight genomic clusters database indicates that glyphosate-tolerant bacteria were more frequently found in soils containing this herbicide compared to soils without glyphosate [39].

## Cell viability in Heat treatment

Cell viability is one of the most important indicators for determining whether cells are experiencing oxidative stress and damage [40]. Herbicides can decrease the cell viability of bacteria, such as *Arthrobacter* sp. with the increase of nicosulfuron [41]. The cell viability of *Stenotrophomonas* sp. CMA26 significantly decreased throughout the growth phases in the 1x treatment, but the opposite was observed for the 50x treatment (Fig 3). This indicates that this strain can differentially modulate the tolerance system based on herbicide concentrations.

The significant increases in cell viability in the mid-phase for the 50x treatment was directly correlated with the period corresponding to the growth recovery of the strain (Fig 1). These data support the hypothesis of decreasing in growth rates as an adaptive period, at least for high concentrations of Heat herbicide. Similar growth recovery behavior was observed in two bacterial strains isolated from water used for wash pesticide containers, in presence of glyphosate [42].

## Indicators of oxidative stress

ROS are products of normal aerobic metabolism, but some factors, such as exposure to ionizing and ultraviolet radiation and to herbicides, can unbalance the production and elimination of ROS, characterizing the oxidative stress [7, 8]. The quantification of $H_2O_2$ and MDA were considered indicators of oxidative stress induced by herbicides, since these substances have toxic elements capable of increasing the levels of peroxide and damage to membrane lipids, or lipid peroxidation [28].

**H2O2 levels in response to Heat.** The amounts of $H_2O_2$ were associated with the growth phases in *Stenotrophomonas* sp. CMA26 (Fig 2). Exposure to high concentrations of Heat herbicide appears to increase $H_2O_2$ production and promote early impacts on cell growth. The

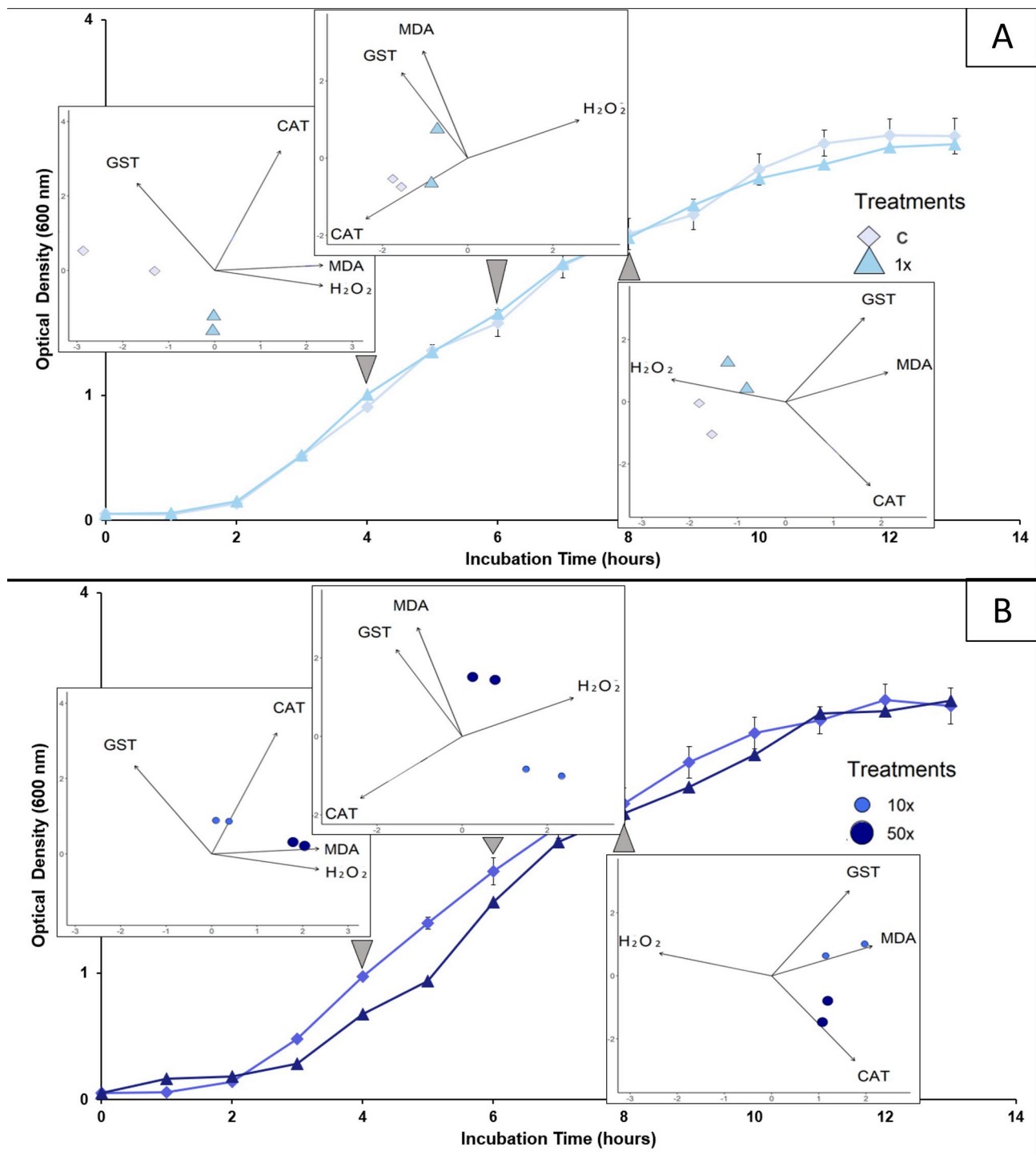

**Fig 7. Integrated panels.** Panels integrate growth curve data and PCA data, according to early, early-mid, and mid phases. Panel A (above) shows the data obtained for control (C) and the Heat herbicide concentration used in the field (1x). Panel B (below) shows the data obtained for 10 and 50x the Heat herbicide concentration used in the field (10x and 50x).

maintenance of higher $H_2O_2$ levels in the 50x treatment in the early and early-mid phases (Fig 2) characterizes concentration as toxic for this strain and may explain the corresponding drop in growth rates (Fig 1). Probably the bacterial response system was efficient in the 10x treatments, despite the high $H_2O_2$ concentrations (Fig 2), because there are no drops in growth rates in this treatment (Fig 1).

By-products of detoxification pathways, changes in electron transport chains, or even inhibition of enzymatic or non-enzymatic antioxidants can increase ROS levels in cells exposed to pesticides [43], impacting, temporarily, the cell growth. For example, in a strain of *E. coli* directly exposed to $H_2O_2$, there was an interruption in cell growth in early periods, due to the deceleration of the synthesis of oxidative defense proteins, mainly those induced by OxyR. Subsequently, translation and growth rates were recovered [44].

**Lipid peroxidation in response to Heat.** MDA is one of the main biomarkers of lipid peroxidation of polyunsaturated fatty acids, and thus it serves as an indicator of oxidative stress [45]. MDA levels in Heat herbicide treatments (Fig 3) are only proportional to $H_2O_2$ levels in the early phase (Fig 2), likely due to the lesser control of lipid peroxidation at this point. Lipids are the most susceptible molecules under oxidative stress. Biological membranes, due to their lipid composition [45], are the cellular components that suffer the most damage. The significant increase in lipid peroxidation, caused by exposure to high concentrations of cadmium and plant-derived compounds such as alpha-pinene and quercetin, was found to be associated with membrane damage in *Rhizobium* sp. [46].

However, the absence of a direct correlation between MDA (Fig 3) and $H_2O_2$ levels (Fig 2) in the early-mid phase suggests changes in membrane lipid saturation in *Stenotrophomonas* sp. CMA26 and a response system to the herbicide. Changes in membrane lipid composition are crucial for bacterial adaptation in response to environmental stress [47]. Different studies suggest that the lack of a direct correlation between MDA and $H_2O_2$ is associated with variations in the saturation of membrane lipids. This is the case of *P. ananatis* [10] and *E. coli* DH5-a [9], in which the increase in lipid saturation was considered as a mechanism for membrane protection through changes in selectivity towards the herbicide mesotrione. Another example is *Pseudomonas fluorescens* CMA-55, which reduces cell membrane permeability by modifying the composition of fatty acids, including nonadecylic acid, margaric acid, and lauric acid. This adaptation allows the bacterium to withstand glyphosate toxicity. This tolerance mechanism was considered independent of prior selection, as glyphosate was not present in biofilms from water storage tanks used to wash herbicide containers, where this strain was isolated [48]. It is possible that this mechanism was used in *Stenotrophomonas* sp. CMA26 from early to early-mid phase. However, lipid saturation returns to lower levels in the mid-phase, possibly due to the modulation of the response, as explained in more detail below.

## Response systems

Response systems enable rapid adaptation to environments that experience intense changes in chemical constituents, such as agricultural soil [27]. The degradation capacity and antioxidant activities of CAT and GST enzymes are potencial response systems of *Stenotrophomonas* sp. CMA26 to Heat herbicide were evaluated in this study.

**Heat degradation as bacterial response.** Prior exposure to herbicides is often reported as a selective agent for degrading microorganisms. Genomic analysis of a microbial community of wastewater from pesticide biopurification systems allowed the identification of genes and gene clusters related to the degradation of organic products, including herbicides, as the linuron, present in environment [49]. *Ochrobactrum* sp., a thifensulfuron-methyl-degrading bacterium, was isolated from soil contaminated with this herbicide [21]. Thus, the absence of

previous exposure to Heat herbicide at the isolation site may have been a factor that limited the selection of specific genes for saflufenacil degradation but was not a precondition for the activity of a general stress response systems responsible for tolerance to this herbicide. Corroborating this hypothesis, general protective mechanisms against ROS in response to exposure to oxidative and osmotic stress and to the antibiotic imipenem was obtained through the expression of 194 genes codifying for efflux pumps, universal stress proteins, and redox enzymes in *P. putida* [50]. *E. coli* strains preadapted to ampicillin showed tolerance to norfloxacin, which was not previously exposed to bacteria, through the stress lag-phase elongation, a generalized adaptive response to antibiotic [51].

The discussion about the bacterial response system can be extrapolated to the soil with a reasonable level of security, since both this environment and the culture medium used to obtain the in vitro results are rich in nutrients. The effect of selecting microorganisms' pesticide-tolerant, but not necessarily degrading, can impact soil diversity, because soil microbiota plays a key role in regulating biogeochemical cycles as nitrogen [52], carbon, nutrient, and many other [53]. Changes in structure of microbiomes in agricultural soil, in response to flufenacet and isoxaflutol herbicides, reduced soil enzyme activities and corn crop yields [54].

The degradation of the herbicide Heat has already been detected in plants as *Egeria densa* and *Pistia stratiotes* [55]. Considering the horizontal gene transfer, it is possible that the degradation genes are present in bacteria. However, did not detected degrading bacteria in this work, disregarding this response system to overcome the stress induced by the herbicide Heat.

**CAT enzymatic activity in response to Heat.** CAT is an antioxidant enzyme responsible for metabolizing $H_2O_2$ into molecular oxygen and water. It plays a key role in bacteria that are exposed to xenobiotics [28]. The results of CAT activity (Fig 4) indicate that this enzyme plays a key role in controlling $H_2O_2$ levels in *Stenotrophomonas* sp. CMA26 treated with Heat herbicide, mainly during the mid phase, showed significant increases in enzyme activity and a consequent decrease in $H_2O_2$ (Fig 2). The role of the CAT enzyme is often described in the control of stress caused by variuos xenobiotics. *Xanthomonas oryzae* pv. *oryzae* and *X. oryzae* pv. *oryzicola* exhibited activation of the transcriptional regulator OxyR in response to the antibiotic phenazine-1-carboxylic acid, which encodes for CAT genes [56]. Increases in CAT activity have been characterized as a mechanism of tolerance to the herbicides metolachlor and acetochlor in isolates of *Enterobacter* that were selected from an environment previously exposed to these herbicides [57]. *Pseudomonas* sp., previously exposed to the Heat herbicide, exhibited a response system that relied on antioxidant enzymes, such as CAT, SOD, and GST [28]. Curiously, there are no reports in the literature demonstrating the role of CAT in conferring tolerance to microorganisms that have not been previously exposed to pesticides. However, the CAT-$H_2O_2$ system alone was insufficient to explain stress control before the mid-log phase, even in the presence of the pesticide. The activity of the GST enzyme was evaluated in response to the herbicide Heat.

**GST enzymatic activity in response to Heat.** GST is a multifunctional enzyme that participates in cellular detoxification processes. It catalyzes the conjugation of GSH with endogenous electrophilic compounds such as secondary metabolites and hydroperoxides as well as xenobiotics [58]. In bacteria, GST is also involved in the degradation of herbicides [59]. Another function of GST that has been studied more recently is the regulation of lipid peroxidation products [45]. MDA is one of the main components of the cell signaling pathway [60]. Here, since Heat degradation was not identified, other GST functions were investigated. The proportionality of the GST activity (Fig 5) and MDA level (Fig 3) suggests that *Stenotrophomonas* sp. CMA26 presents a system for regulating the levels of peroxidation through the activity of this enzyme during the early-mid phase. Nevertheless, few papers address the relationship between GST and MDA. GST was found to be involved in the regulation of stress caused by

mesotrione through the degradation of the herbicide via GST-mesotrione conjugates in *P. ananatis* [10]. In this case, the degradation of the molecule likely result in the formation of free thiol groups. These groups were expected to induce lipid peroxidation, leading to an increase in MDA levels. But this effect was not observed, and peroxidation control was not related to GST for this bacterial strain.

GST may involved in the elimination of lipids damaged by $H_2O_2$ and the restoration of membrane fluidity in microorganisms that are exposed to temperature stress [61]. GST could catalyze the degradation of lipid peroxidation products in the biological membranes of *Rhizobium* sp., which were collected from non-contaminated soil with cadmium [46]. GST was associated with the regulation of MDA levels in *Pseudomonas* sp. in response to stress caused by the herbicide Boral. This strain of *Pseudomonas* was obtained from a location that had not been previously exposed to herbicide [28]. Therefore, GST-MDA can play a key role in protecting biological membranes and is part of a system that is independent of previous selection. As was questioned for the CAT-$H_2O_2$ system, the GST-MDA system did not respond to the stress control generated by the Heat herbicide from the mid-log phase. This lead to an integrated analysis of these two systems in search of a more consistent hypothesis to explain the response system of *Stenotrophomonas* sp. CMA26.

**Integrated response systems for Heat tolerance.** Two enzymatic response systems were found in *Stenotrophomonas* sp. CMA26, one modulated according to the growth phase, and the other dependent on herbicide concentration, allowed the survival of this strain in the presence of the herbicide Heat *in vitro*, even without previous selective pressure *in situ*.

The first response system can be identified through the PCA data (Fig 6), which shows the interrelationships among $H_2O_2$, MDA quantifications, and the activities of CAT and GST enzymes at each growth phase. This suggests a modulation of the antioxidant response of the *Stenotrophomonas* sp. CMA26 is resistant to the herbicide Heat. The PCA provided a clearer understanding of the relationships between the various data collected, such as the relationship between $H_2O_2$ and MDA in the early phase under the 50x treatment. This suggests that this concentration was stressful for the strain, possibly due the peroxide-peroxidation. The negative effects on growth in this condition support this hypothesis (Fig 1). Additionally, there is no significant enzymatic activity involved in the regulation of $H_2O_2$ and MDA stress markers during this phase. However, the distribution of CAT and $H_2O_2$ in opposite quadrants during the early-mid and mid phases characterizes the role of CAT in controlling $H_2O_2$. In the same phases, the distribution of GST and MDA in the same quadrants indicates that the amount of MDA induces GST activity. Heat herbicide induces early-stage stress condition in *Stenotrophomonas* sp. CMA26, which is overcome during the early-mid phase, involves the activities of the CAT and GST enzymes.

The modulation in the metabolic response to tolerate $H_2O_2$ in *Staphylococcus aureus*, observed by amino acid composition profiles, differed according to the incubation temperature and the pH of the culture medium [62]. Here, metabolic homeostasis would be continuously adjusting and responding to changes in culture conditions, including the expression of antioxidant proteins, for the survival of *S. aureus* against $H_2O_2$. Likewise, *Stenotrophomonas* sp. CMA26 could be continuously regulating the metabolic system to respond to the stress caused by Heat herbicide, characterizing a system of metabolic plasticity. Bacterial metabolic plasticity is the ability to respond through changes in morphology, physiology, or activity to tolerate environmental stressors [63]. *Stenotrophomonas* exhibits several strategies for adaptation to different environmental niches, as plastic metabolic systems [14], xenobiotic degradation activities [64], biofilm formation [65], quorum sensing and quorum quenching [66].

The performance of this plastic and modulated system may have occurred in the 50x treatment in *Stenotrophomonas* sp. CMA26, with a decrease in the growth rate in relation to the

other treatments, followed by recovery. The bacterial adaptation to new environments is strongly influenced by previous exposure to stressing agents [67]. Bacteria pre-adapted to stressful conditions, for example, have shorter growth inhibition periods than non-pre-adapted bacteria in the same stress conditions [68]. This inhibition usually occurs in the latency (lag) phase, which is commonly described as a period of no growth, but it is a dynamic, organized, adaptive and evolutionary process that protects bacteria from threats and promotes reproductive fitness [69]. The prolongation of the lag phase is a defense mechanism that allows bacteria to tolerate stress, including for substances to which they have not been previously exposed; also, tolerance can be achieved when growth-inhibited cells have lower metabolic activity than actively dividing cells [68]. No articles were found discussing adaptation during the log phases; however, the lag and early phases of the log are metabolically connected. Thus, the early phase may be a period of metabolic reduction and gene activation in that confer Heat herbicide tolerance, at least for high concentrations of herbicide.

The second response system, related to herbicide concentration, can be identified by changes in the clustering patterns among the quadrants of the main components (Fig 6). These changes are reflected in two modes of modulation of the activities of the analyzed enzymes: one for the control and 1x treatments, and another for the 10x and 50x treatments (Fig 7). One mode has been found for controlling the lowest concentration of herbicide, in which the bacterium appears to influence energy production metabolism through CAT activity. The pattern changed for higher herbicide concentrations, which were associated with higher stress condition. This can lead to damage to the bacterial membrane, as indicated by the association of these treatments with stress indicators, particularly MDA. These treatments required a higher activity of the GST enzyme to overcome these conditions in the mid-phase. For all herbicide concentrations tested in this study, the adaptive capacity of *Stenotrophomonas* sp. CMA26 was maintained, as can be observed by the growth curves (Fig 1), indicating the fitness of this response-modulated system to the herbicide Heat.

GST, from the sigma class of *Phlebotomus argentipes*, is responsible for conferring tolerance to DDT by acting against 4-hydroxy-nonenal (4-HNE), a byproduct of the peroxidation of long-chain lipid hydroperoxides [70]. In *Arabis alpina*, a GST mitigates the damage caused by lipid peroxidation due to exposure to ultraviolet radiation [71]. GST is present in several taxonomic groups, from prokaryotes to eukaryotes, as it has functions related to the control of lipid peroxidation. Therefore, it is possible that GST is also present in *Stenotrophomonas* sp. CMA26 is also included, according to the data obtained in this study.

This ROS response system may have been found in *Stenotrophomonas* sp. CMA26 modulates the activities of various antioxidant enzymes, regardless of their origin, to enhance survival in environments with frequent exposure to new xenobiotics, such as agricultural soils. Degradation is a process that depends on specific genes and prior selection by the herbicide. For example, the degradation of 2,4-dinitrotoluene (DNT) in *Burkholderia* sp. The occurrence is due to the presence of the *dnt* gene [72]. However, the degradation pathway of the compound leads to the excessive production of endogenous ROS in cells. This, in turn, causes DNA mutagenesis and selects for a subset of the population that is tolerant. In another example, the herbicide paraquat induced the expression of the *sodB* gene, which encodes SOD in the root endophyte *B. seminalis* TC3.4.2R3 [73]. Here, the herbicide acted as a non-specific stressors, as SOD catalyzes $O_2^-$, favoring this genotype and phenotype. From there, the selected metabolic system would be ready to act on other xenobiotics or metabolic conditions that cause oxidative stress, thereby increasing the adaptive value of the species. ROS can be produced through normal cellular metabolic processes, as well as in response to radiation exposure and pesticides such as herbicides [8].

Loss or gain of soil functions may be an effect of the selection of plastic phenotypes. The herbicide glyphosate, which is similar to the Heat herbicide in terms of application, enhanced the growth of two strains of Firmicutes and *Burkholderia*. This strains inhibit root growth in a synthetic bacterial community that imitates the microbiome associated with the *Arabidopsis thaliana* root [74]. Glyphosate also reduced soil fungal biomass by 29%, but it subsequently increased the catabolic activity of gram-negative bacteria [75]. These mechanisms of metabolic plasticity, which result in herbicide tolerance regardless of herbicide degradation, may also be present in other soil microorganisms. This was assessed for the other isolates in the soil collection during this study. In this sense, herbicides can persist in the soil as environmental contaminants for extended periods, increasing the likelihood of reaching aquatic environments through drift and affecting the soil microbiome. Environmental contamination caused by herbicides is an undeniable fact.

## Conclusions

Agricultural soils are ecosystems that undergo to intense environmental changes, including fluctuations in nutrients and water availability, as well to the application of pesticides. An important question to understand the dynamics of fertility and soil impacts is how selective pressure can shape the response system of microorganisms in these environments. *Stenotrophomonas* sp. CMA26, a strain of agricultural rhizospheric soil, showed tolerance to the herbicide Heat, even when isolated from agricultural soil without *in situ* exposure to this xenobiotic. This bacterial strain exhibited a regulated response system during various growth phases. Bacterial metabolism displayed specific sensitivities to oxidative stress induced by Heat herbicide, with the GST enzyme regulating MDA levels during the early stages of growth, while the CAT enzyme controlled $H_2O_2$ levels in the later stages. A modulation system was observed these enzymatic systems in response to different herbicide concentrations. At lower the system was associated with CAT activity and $H_2O_2$ production. However, at higher concentrations, the system showed a greater association with CAT and GST activities and MDA production. This suggests that at the highest concentrations, the level of lipid peroxidation could be more harmful to the bacterial cell. This system of enzymatic responses could be evaluated at *in vitro* concentrations considerably higher than those found in soils used in agriculture. This demonstrates its effective ability to control stress, even for xenobiotics that are not present in these environments.

These types of adaptive mechanisms may be found in other soil microorganisms, as other isolates from the collection exhibited Heat herbicide tolerance without showing herbicide degradation. Soil microbiomes that do not rely on previous selection mechanisms can experience disruptions in their diversity and functionality, which can ultimately affect biogeochemical cycles. Since herbicide degradation is not the only bacterial survival strategy, these xenobiotics may remain in the soil for longer periods, increasing the likelihood of reaching other environmental compartments through drift. Despite scientific advances in agriculture and environmental issues, the planet is experiencing increasing pollution and impact.

## Supporting information

**S1 Appendix. Experimental design.** The experimental design indicates the hierarchical order (left to right) of the experiments performed in this article. The microorganisms collected from agricultural soil were submitted to a tolerance test for different concentrations of Heat. The results allowed selecting the strain, identified as *Stenotrophomonas* sp. CMA 26, which underwent stress indicator tests (growth curve, MDA and $H_2O_2$). The response systems degradation

and activities of CAT and GST enzymes were evaluated.
(DOCX)

**S2 Appendix. Transept localization.** Sample collection points per transept, obtained by Global Positioning System (GPS). M7, M8, M9 and M10 represent soybean and corn crop areas. R1, R2, R3, and R4 represent the parallel transepts traced for the collection.
(DOCX)

**S3 Appendix. Chemical characterization of herbicides.** Pesticides used in the collection area. Trademarks of herbicides, fungicides and insecticides and their respective active molecules and chemical structures, modes of action, chemical families, and classification of chemical families according to the Herbicide Resistance Action Committee (HRAC), Fungicide (FRAC) and Insecticide (IRAC) are presented, in addition to the type of culture in which they were used.
(DOCX)

**S4 Appendix. Statistical analysis.**
(DOCX)

**S5 Appendix. Cell viability.** Cell viability (in CFU) of *Stenotrophomonas* sp. CMA26 in control (C) and treatments containing 1x, 10x, and 50x the concentration equivalent to that used in the field of herbicide Heat (1x, 10x, and 50x), in the early, early-mid and mid phases of the log. Uppercase letters statistically compare different treatments from the same growth phase; lowercase letters statistically compare the same treatments at different growth stages. The bars represent the standard errors in the means. Tukey's test ($p < 0.05$).
(DOCX)

## Acknowledgments

The authors thank Traudi Klein for assistance with the spectrometric analysis.

## Author Contributions

**Conceptualization:** Caroline Rosa Silva, Paloma Nathane Nunes de Freitas, Luiz Ricardo Olchanheski, Marcos Pileggi.

**Data curation:** Caroline Rosa Silva, Paloma Nathane Nunes de Freitas, Luiz Ricardo Olchanheski, Marcos Pileggi.

**Formal analysis:** Caroline Rosa Silva, Luiz Ricardo Olchanheski, Marcos Pileggi.

**Funding acquisition:** Marcos Pileggi.

**Investigation:** Caroline Rosa Silva, Luiz Ricardo Olchanheski, Luciana Grange, Marcos Pileggi.

**Methodology:** Caroline Rosa Silva, Amanda Flávia da Silva Rovida, Juliane Gabriele Martins, Paloma Nathane Nunes de Freitas, Luiz Ricardo Olchanheski, Luciana Grange, Marcos Pileggi.

**Project administration:** Luiz Ricardo Olchanheski, Marcos Pileggi.

**Resources:** Luiz Ricardo Olchanheski, Marcos Pileggi.

**Software:** Luiz Ricardo Olchanheski.

**Supervision:** Luiz Ricardo Olchanheski, Sônia Alvim Veiga Pileggi, Marcos Pileggi.

**Validation:** Caroline Rosa Silva, Paloma Nathane Nunes de Freitas, Luiz Ricardo Olchanheski, Marcos Pileggi.

**Visualization:** Caroline Rosa Silva, Paloma Nathane Nunes de Freitas, Luiz Ricardo Olchanheski, Marcos Pileggi.

**Writing – original draft:** Caroline Rosa Silva, Paloma Nathane Nunes de Freitas, Luiz Ricardo Olchanheski, Luciana Grange, Sônia Alvim Veiga Pileggi, Marcos Pileggi.

**Writing – review & editing:** Caroline Rosa Silva, Amanda Flávia da Silva Rovida, Juliane Gabriele Martins, Paloma Nathane Nunes de Freitas, Luiz Ricardo Olchanheski, Sônia Alvim Veiga Pileggi, Marcos Pileggi.

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
