## [Decision Letter · Decision Letter 0]

21 Jun 2023

PONE-D-23-12907Bacteria adaptation to rhizosphere soil is independent of the selective pressure exerted by the herbicide saflufenacil, through the modulation of Catalase and Glutathione S-transferasePLOS ONE

Dear Dr. Pileggi,

Thank you for submitting your manuscript to PLOS ONE. After careful consideration, we feel that it has merit but does not fully meet PLOS ONE’s publication criteria as it currently stands. Therefore, we invite you to submit a revised version of the manuscript that addresses the points raised during the review process.

We look forward to receiving your revised manuscript.

Kind regards,

Eugenio Llorens

Academic Editor

PLOS ONE

Reviewers' comments:

Reviewer's Responses to Questions

**Comments to the Author**

1. Is the manuscript technically sound, and do the data support the conclusions?

Reviewer #1: Partly

Reviewer #2: Yes

2. Has the statistical analysis been performed appropriately and rigorously? 

Reviewer #1: Yes

Reviewer #2: No

3. Have the authors made all data underlying the findings in their manuscript fully available?

Reviewer #1: Yes

Reviewer #2: Yes

4. Is the manuscript presented in an intelligible fashion and written in standard English?

Reviewer #1: No

Reviewer #2: Yes

5. Review Comments to the Author

Reviewer #1: This study mainly evaluated the tolerance of bacteria Stenotrophomonas sp. CMA26 on herbicide Heat. The authors found that Bacteria CMA26 adaptation is independent of the selective pressure exerted by the herbicide saflufenacil, through the modulation of Catalase and Glutathione S-transferase. The results have some significances for understanding the effects of pesticides on soil microorganisms.

Specific comments

1. The language needs to be further improved. The article doesn't read smoothly.

2. Abstract Ln32-34, the sentence should be deleted from the abstract. Abstract is not discussion.

3. Ln 58, what is Heat? Is it a pesticide commercial name? If yes, give a standard name.

4. Ln 165, where are the section 2.2 and 2.1?

5. Conclusion, please give a clear conclusion. I don't know what the purpose of this work according to the article.

Reviewer #2: Dear Editor,

The current submission deals with the tolerance mechanism of soil-isolated bacteria to HEAT herbicide. Generally, the language of the MS needs to be revised; most of the sentence structures require careful revision. The long title is fine and informative; the short one may be deleted.

Abstract:

Lines 27–29: Some sentences are repeated.

Line 42: What does the author mean by ".. originated them..."?

Introduction:

It is OK. However, details on different mechanisms of bacterial tolerance to herbicides should be provided, and references should be updated.

Material and Methods:

Suitable methods were applied, but other methods (e.g., gene expression) were needed to uncover the exact mechanism for HEAT herbicide tolerance.

Discussion:

It looks like results; the author should really discuss their findings.

Conclusion:

OK

References:

needs an update with recent references.

6. PLOS authors have the option to publish the peer review history of their article (what does this mean?). If published, this will include your full peer review and any attached files.

Reviewer #1: No

Reviewer #2: No

---

## [Author Response · Author response to Decision Letter 0]

5 Aug 2023

Response to Reviewers

Dear Dr.

Eugenio Llorens

Academic Editor

PLOS ONE

Thank you to the editor and reviewers for evaluating and considering improvements to the submitted manuscript. Please find attached our revised research article entitled “Bacterial Adaptation to Rhizosphere Soil is Independent of the Selective Pressure Exerted by the Herbicide Saflufenacil, through the Modulation of Catalase and Glutathione S-transferase.”

We are submitting a letter that responds to each point raised by the reviewers, the marked-up copy of the revised manuscript with track changes made to the original version, and an unmarked version of the revised paper without tracked changes.

Sincerely, the authors.

Authors' response: Modifications were made to the manuscript to comply with the style requirements of the journal, as specified in the provide manuals.

Authors' response: No permits were required to carry out this work. However, we have added information about the collection of agricultural soil, indicating the person responsible, and have included her as the author. Revised Text: Lines 220–222 of the Revised Manuscript with Track Changes.

Authors' response: There was no grant number. The development agency granted a scholarship to the author, Caroline Rosa Silva. Changes have been made to the “Funding Information” and “Financial Disclosure” sections to ensure that the correct information is provided. In the manuscript, the “Funding Information” section has also been added to explain the scholarship grant. Revised Text: Lines 991–993 of the Revised Manuscript with Track Changes.

Authors' response: The phrase has been removed. Revied Text: Line 470 of the Revised Manuscript with Track Changes. We believe that presenting a negative result for herbicide degradation in the manuscript would be of little relevance.

Authors' response: A "Supporting Information" section has been added to the manuscript. In this section, captions were included for the respective files. Revised Text: Lines 1498–1522 of the Revised Manuscript with Track Changes.

Reviewer #1:

Reviewer #1: This study mainly evaluated the tolerance of bacteria Stenotrophomonas sp. CMA26 on herbicide Heat. The authors found that Bacteria CMA26 adaptation is independent of the selective pressure exerted by the herbicide saflufenacil, through the modulation of Catalase and Glutathione S-transferase. The results have some significances for understanding the effects of pesticides on soil microorganisms.

Authors' response: Thank you very much for reviewing the manuscript. We strive to address all the concerns raised by the reviewer.

1. The language needs to be further improved. The article doesn't read smoothly.

Authors' response: The manuscript was revised using the "WORDVICE AI" platform to improve the readability of the text.

2. Abstract Ln32-34, the sentence should be deleted from the abstract. Abstract is not discussion.

Authors' response: The sentence has been deleted, and the abstract has been rewritten. Revised Text: Lines 34–58 of the Revised Manuscript with Track Changes.

3. Ln 58, what is Heat? Is it a pesticide commercial name? If yes, give a standard name.

Authors' response: Yes, Heat is a commercial brand of herbicide, in Brazil that contains saflufenacil as its active ingredient. We have made modifications to the manuscript in order to clarify the definition in the introduction (Lines 99–100 of the Revised Manuscript with Track Changes) and throughout the entire text.

4. Ln 165, where are the section 2.2 and 2.1?

Authors' response: We excluded this sentence from the manuscript because the format does not have numbered sections. Revised Text: Lines 217, 224, 226, 253, 262, 269, and 336 of the Revised Manuscript with Track Changes.

5. Conclusion, please give a clear conclusion. I don't know what the purpose of this work according to the article.

Authors' response: Thank you very much for your review. The conclusions were rewritten to address the objectives of the study, which were to assess the presence of a response system to tolerate Heat herbicide in vitro, but not in situ, and to examine the implications of this response system on microbial ecology and soil contamination. Revised Text: Lines 917–944 of the Revised Manuscript with Track Changes.

Reviewer #2:

Reviewer #2: Dear Editor, 

The current submission deals with the tolerance mechanism of soil-isolated bacteria to HEAT herbicide. Generally, the language of the MS needs to be revised; most of the sentence structures require careful revision. The long title is fine and informative; the short one may be deleted.

Authors' response: Thank you very much for reviewing the manuscript. We strive to address all the concerns raised by the reviewer. The manuscript was revised by the "WORDVICE AI" platform to improve the language structure. The short title has been deleted. Revised Text: Line 7 of the Revised Manuscript with Track Changes.

Abstract:

Lines 27–29: Some sentences are repeated. 

Line 42: What does the author mean by ".. originated them..."?

Authors' response: We excluded repeated sentences. The abstract has been rewritten. We would like to clarify that this response system, which is capable of tolerating heat regardless of previous selection for this herbicide, may have evolved in response to reactive oxygen species, regardless of the substance causing the oxidative stress. Revised Text: Lines 52–54 of the Revised Manuscript with Track Changes.

Introduction:

It is OK. However, details on different mechanisms of bacterial tolerance to herbicides should be provided, and references should be updated.

Authors' response: Details on the various mechanisms of bacterial tolerance have been included. Revised Text: Lines 158–159, 159–160, 162, and 163 of the Revised Manuscript with Track Changes. We have also updated the references in the Introduction. Revised Text: Lines 96, 121-122, 131-132, 142, 158–161 of the Revised Manuscript with Track Changes.

Material and Methods:

Suitable methods were applied, but other methods (e.g., gene expression) were needed to uncover the exact mechanism for HEAT herbicide tolerance.

Authors' response: Thank you very much for the suggestion. Methods that include gene expression analysis will undoubtedly enhance data on response systems. However, the intention of this work was to initiate the characterization of a response system by describing the performance of two enzymes, CAT and GST. Our future work will include gene expression analysis to further enhance the robustness of the response system.

Discussion:

It looks like results; the author should really discuss their findings.

Authors' response: The discussion has been expanded. Revised Text: Lines 652–681, 756–757, 760–767, 779–783, 787-790, 853–876, 877–883 of the Revised Manuscript with Track Changes. Fig. 7, which was previously located in the "Discussion" section, has been moved to the "Results" section. Revised Text: Lines 539–542 of the Revised Manuscript with Track Changes.

Conclusion:

OK

Authors' response: Thank you very much for your review.

References:

needs an update with recent references.

Authors' response: The references have been updated. The new references included are in lines 1025–1027, 1031–1033, 1062–1065, 1066–1068, 1140–1142, 1152–1155, 1176–1178, 1190–1192, 1201–1203, 1212–1214, 1215–1219 of the Revised Manuscript with Track Changes.

---

## [Decision Letter · Decision Letter 1]

3 Oct 2023

Bacterial adaptation to rhizosphere soil is independent of the selective pressure exerted by the herbicide saflufenacil, through the modulation of catalase and glutathione S-transferase

PONE-D-23-12907R1

Dear Dr. Pileggi,

We’re pleased to inform you that your manuscript has been judged scientifically suitable for publication and will be formally accepted for publication once it meets all outstanding technical requirements.

Kind regards,

Eugenio Llorens

Academic Editor

PLOS ONE

Additional Editor Comments :

During the last round of review, one of the reviewers found a mistake in the primers of 16S. Please, be sure to correct it during the editing of the manuscript.

Reviewers' comments:

Reviewer's Responses to Questions

**Comments to the Author**

1. If the authors have adequately addressed your comments raised in a previous round of review and you feel that this manuscript is now acceptable for publication, you may indicate that here to bypass the “Comments to the Author” section, enter your conflict of interest statement in the “Confidential to Editor” section, and submit your "Accept" recommendation.

Reviewer #1: (No Response)

Reviewer #2: All comments have been addressed

2. Is the manuscript technically sound, and do the data support the conclusions?

Reviewer #1: (No Response)

Reviewer #2: Yes

3. Has the statistical analysis been performed appropriately and rigorously? 

Reviewer #1: (No Response)

Reviewer #2: Yes

4. Have the authors made all data underlying the findings in their manuscript fully available?

Reviewer #1: (No Response)

Reviewer #2: Yes

5. Is the manuscript presented in an intelligible fashion and written in standard English?

Reviewer #1: (No Response)

Reviewer #2: Yes

6. Review Comments to the Author

Reviewer #1: (No Response)

Reviewer #2: (No Response)

7. PLOS authors have the option to publish the peer review history of their article (what does this mean?). If published, this will include your full peer review and any attached files.

Reviewer #1: No

Reviewer #2: No

---

## [Editor Report · Acceptance letter]

6 Nov 2023

PONE-D-23-12907R1 

Bacterial adaptation to rhizosphere soil is independent of the selective pressure exerted by the herbicide saflufenacil, through the modulation of catalase and glutathione S-transferase 

Dear Dr. Pileggi:

I'm pleased to inform you that your manuscript has been deemed suitable for publication in PLOS ONE. Congratulations! Your manuscript is now with our production department. 

Kind regards, 

on behalf of

Dr. Eugenio Llorens 

Academic Editor

PLOS ONE